# A Dual-Mode 303-Megaframes-per-Second Charge-Domain Time-Compressive Computational CMOS Image Sensor

**DOI:** 10.3390/s22051953

**Published:** 2022-03-02

**Authors:** Keiichiro Kagawa, Masaya Horio, Anh Ngoc Pham, Thoriq Ibrahim, Shin-ichiro Okihara, Tatsuki Furuhashi, Taishi Takasawa, Keita Yasutomi, Shoji Kawahito, Hajime Nagahara

**Affiliations:** 1Research Institute of Electronics, Shizuoka University, Hamamatsu 432-8011, Japan; ttakasawa@idl.rie.shizuoka.ac.jp (T.T.); kyasu@idl.rie.shizuoka.ac.jp (K.Y.); kawahito@idl.rie.shizuoka.ac.jp (S.K.); 2Graduate School of Integrated Science and Technology, Shizuoka University, Hamamatsu 432-8011, Japan; mhori@idl.rie.shizuoka.ac.jp (M.H.); pham.ngoc.anh.17@shizuoka.ac.jp (A.N.P.); thoriq.ibrahim.16@shizuoka.ac.jp (T.I.); tfuru@idl.rie.shizuoka.ac.jp (T.F.); 3Photonics for Material Processing, The Graduate School for the Creation of New Photonics Industries, Hamamatsu 431-1202, Japan; s.okihara@gpi.ac.jp; 4Institute of Datability Science, Osaka University, Suita 565-0871, Japan; nagahara@ids.osaka-u.ac.jp

**Keywords:** ultra-high-speed imaging, multi-tap CMOS image sensor, charge modulator, compressive imaging, computational imaging

## Abstract

An ultra-high-speed computational CMOS image sensor with a burst frame rate of 303 megaframes per second, which is the fastest among the solid-state image sensors, to our knowledge, is demonstrated. This image sensor is compatible with ordinary single-aperture lenses and can operate in dual modes, such as single-event filming mode or multi-exposure imaging mode, by reconfiguring the number of exposure cycles. To realize this frame rate, the charge modulator drivers were adequately designed to suppress the peak driving current taking advantage of the operational constraint of the multi-tap charge modulator. The pixel array is composed of macropixels with 2 × 2 4-tap subpixels. Because temporal compressive sensing is performed in the charge domain without any analog circuit, ultrafast frame rates, small pixel size, low noise, and low power consumption are achieved. In the experiments, single-event imaging of plasma emission in laser processing and multi-exposure transient imaging of light reflections to extend the depth range and to decompose multiple reflections for time-of-flight (TOF) depth imaging with a compression ratio of 8× were demonstrated. Time-resolved images similar to those obtained by the direct-type TOF were reproduced in a single shot, while the charge modulator for the indirect TOF was utilized.

## 1. Introduction

Ultra-high-speed (UHS) cameras are utilized to observe ultrafast phenomena in the fields of science and industry [1]. Solid-state UHS cameras are advantageous compared with other technologies based on streak cameras or ultra-short pulse lasers [2,3] in terms of compactness, durability, and mass productivity, although increasing the frame rate is a challenge. To realize ultrafast frame rates that cannot be achieved by ordinary continuous-readout image sensors, the burst readout scheme is utilized. Burst readout UHS image sensors are equipped with on-chip frame memory, where images are stored for a short time. There are several implementations, such as CCD or CMOS, with pixelwise or column-wise memory. UHS image sensors with on-chip analog frame memory have achieved 20 megaframes per second (Mfps)) [4]. UHS CMOS image sensors with in-pixel memory are more suitable for high frame rates, and have demonstrated burst image acquisition at over 100 Mfps [5,6]. The maximum frame rate is limited by the time to write a pixel value into the frame memory or to operate the CCD registers with an acceptable charge transfer efficiency.

We proposed a multi-aperture UHS image sensor that adopted an entirely different scheme using compressive sensing, which was motivated by computational photography [7,8]. Compressive video is a similar technique, although the frame rates are close to video rates, namely, 30 fps [9,10]. In this scheme, input optical images are compressed in the charge domain as an inner-product or correlation with a random, temporally coded shutter at each pixel. Because the pixels themselves work as the frame memory, no additional frame memory or signal processing circuit on the chip is required. The most important advantage is that the burst frame rate is determined only by the speed of one-step charge transfer from the photodiode to the storage diode. Furthermore, this image sensor can be used for both multi-exposure imaging, such as time-of-flight (TOF) range imaging, and single-event filming, whereas conventional UHS image sensors can implement either of these two filming modes. The key device in this image sensor is a charge modulator, which is mainly used in indirect time-of-flight image sensors [11]. For example, charge modulators based on photogate [12], transfer gate [13,14], and lateral electric field charge modulator (LEFM) [15] have been demonstrated. A single-electron avalanche diode (SPAD) [16] is the core device of direct TOF, and it cannot be used in single-event ultrafast imaging because of the limitation of the photon rate, which is determined by the dead time and shared time-to-digital converters. The charge modulator is suitable for the compression of UHS images because low-level time-domain signal processing, i.e., the inner-product or correlation operation between an incident light signal and an electric gating signal, is performed in the charge domain. However, one of the drawbacks of a multi-aperture UHS image sensor is that it requires unconventional multi-aperture optics, making it difficult to combine this image sensor with traditional single-aperture lenses. This limitation can be overcome by adopting a macro-pixel-based architecture [17], which is compatible with conventional lenses. However, the design issues of the pixel driver limit the burst frame rate to 73 Mfps.

This paper presents the fastest silicon UHS image sensor that utilizes charge-modulator-based macropixels with 2 × 2 4-tap subpixels (16 taps per macropixel). The burst frame rate is 303 Mfps. To achieve this frame rate, the pixel drivers were adequately designed to suppress the peak current, taking advantage of the driving constraint of the multi-tap charge modulators. In total, 32 images are reproduced from 4 compressed mosaic images captured in a single shot. This image sensor can operate in dual modes, such as single-event mode or multi-exposure mode, selectively, although conventional UHS image sensors cannot used as TOF image sensors and vice versa. In Section 2, the fundamental procedures of image acquisition and signal processing are shown. In Section 3, the implementation of a new image sensor is described. In Section 4, the fabricated image sensor is characterized and temporally compressive filming in single-event and multi-exposure modes is demonstrated. In Section 5, some issues with the sensor and future work are discussed.

## 2. Macro-Pixel Multi-Tap Computational CMOS Image Sensor Architecture

### 2.1. Imaging System Configuration

Figure 1 shows an overview of the image sensor and imaging system. The image sensor has an array of macropixels. They are composed of 2 × 2 4-tap subpixels, each of which has a photodiode and four storage diodes. Charge transfer is controlled by a temporally coded shutter. First, an optical image of the object is formed on the image sensor by a single-aperture lens with a point spread function (PSF) that can cover the whole macropixel. In filming, an inner-product, or correlation, of the optical images and a temporally-coded shutter for each subpixel and tap is stored in the storage diode of each tap. Next, four temporally compressed mosaic images or inner-product images are read out. Finally, a series of original images is reproduced by solving an inverse problem on a computer with a compressive sensing solver with the PSF and the temporal impulse response of the taps measured beforehand.

### 2.2. Compressive Sensing and Imaging

Image acquisition and reproduction are based on compressive sensing [18,19,20,21]. The mathematical representation of compressive sensing is briefly mentioned here. When we consider an N-dimensional column vector x, an M×N matrix A, and an M-dimensional column vector y, the relationship is denoted by
(1)y=Ax.

Note that x is an original input signal, and y is the signal measured through the measurement matrix A, which is known. When N>M, the original signal x is compressed. We can reproduce the original input signal x by solving the inverse problem from y and A with a sparsity constraint. 

The image sensor response, applied coded shutters, and PSF of the imaging lens are represented by the single measurement matrix A. When we think of a single point measurement case for simplicity, it is assumed that A is composed of M sets of temporal shutters given by an M×N-dimensional matrix. Note that N is the shutter length or the number of input images, and M is the number of compressed images. Note that in the case of two-dimensional imaging with image sensors, the dimensions of N and M are multiplied by the number of pixels of the image sensor. The number of total taps is as many as that of the taps in the subpixel times that of the subpixels in the macro-pixel, namely, M = 4 × 4 = 16. When the number of the pixels for one reproduced image is equal to that of the subpixels of the image sensor, the compression ratio is defined by the ratio of the number of the reproduced images to that of the taps in each subpixel.

We assume that the original input signal x is K-sparse, which means that only K elements have non-zero values and all the other elements are zero. Total variation (TV) minimization is widely used to solve $l_{1}$-norm minimization problems. The original input signal x is estimated as
(2)$${\hat{x}}^{\left(TV\right)}=\underset{x}{\mathrm{argmin}}\Sigma_{i}∥D_{i}x{∥}_{1}\;\mathrm{subject}\;\mathrm{to}\;y=Ax,$$
where $D_{i}$ is the differential operator used to subtract an adjacent element from the i-th element.

Applications of compressive sensing have been widely studied in a variety of fields, such as video capturing [22], multi-spectral imaging [23], astronomy [24], medicine [25], security [26], manufacturing [27], and so on. Dedicated spatio-temporal compressive CMOS image sensors have been developed for efficient image acquisition with low power consumption, higher frame rates, or higher pixel count. Articles on compressive image sensors based on the image compression by analog or digital circuits are reviewed in [28]. The authors of [9,10] adopt a different signal compression scheme where the signal is compressed in the charge domain in pixel. This scheme is advantageous in temporal compression in terms of pixel size and power consumption. Our method is also based on charge domain signal compression to achieve ultra-high frame rates, as described in Section 1.

## 3. Chip Implementation

### 3.1. Sensor Architecture

The sensor architecture is shown in Figure 2. There are two major issues in the circuit design of the UHS computational CMOS image sensor. Firstly, to realize a high operational frequency, which is the same as the burst frame rate, the portion of the shutter controller circuits operating at the maximum frequency should be minimized. This operation scheme is also effective at reducing the power consumption. Secondly, for the charge modulator drivers, the suppression of the peak driving current is key. Because the drivers share the same power and ground lines, the number of drivers that operate at the same time should be reduced. Otherwise, the voltage drop in the shared lines will cause a significant performance degradation in the charge modulation. These issues are solved by the circuit architecture design, as mentioned below.

The shutter controller (SC) stores shutter patterns and provides them to the charge modulator drivers. The start and stop timings of filming are controlled by the external signal, TRG. The shutter patterns are denoted by GSPx[4:1] (x: subpixel index, number in the brackets: tap index, as shown in Figure 1 and Figure 3. The shutter pattern memory stores 1 to 32 8-bit patterns for each of the 16 taps, which are programmed via SPI-1 (SPI: serial to parallel interface). The shutter pattern length, phase lock loop (PLL) frequency, and other image sensor operation parameters are programmed via SPI-2. As described below, tap-4 can work as a charge drain when ENDRN is asserted. During the capture-ready state, the drain controller always sets GSPx[4] to H.

### 3.2. Pixels and Drivers

Figure 3 shows the architecture of the charge modulator drivers and macropixel array, which reduces the peak driving current of the drivers at a high operational frequency by taking advantage of the driving constraint of the multi-tap charge modulator. One of the most significant design issues of the charge modulator driver is an IR voltage drop on the power lines, since huge charge/discharge and transient currents for the whole pixel array flow into them. In this design, power lines are separately prepared for each subpixel. Note that for multi-tap charge modulators, it is allowed to turn on only one tap for a multi-tap charge modulator, since the amount of charge transferred to the taps becomes uncertain when multiple taps turn on at the same time. Taking advantage of this operational rule, the maximum current that flows into the power lines can be limited. In the architecture shown in Figure 3, the Hamming distance of the shutter pattern for a subpixel between the adjacent timings is necessarily zero or four (not more). A subpixel circuit is also shown in Figure 3. The charge modulator is based on the LEFM shown in [29]. The charges are stored at FDs. The value j indicates a row position (j = 0, 1, …). Because tap-4 can work as a drain, a reset signal RS4 <2j + 1> is separately prepared. When ENDRN is H, RS4 <2j + 1> always becomes H to drain the charges transferred to tap-4.

### 3.3. Timing Chart

This image sensor can be set to operate in dual modes, such as single-event mode or multi-exposure mode, selectively, by reconfiguring the number of exposure cycles. Figure 4 shows example waveforms of the control signals. Filming begins when TRG = H is detected at the negative edge of CLK. The other parts of SC work at the positive edge of CLK. Next, RDY, which indicates capture-ready, turns to L, and the shutter patterns are read out from memory. In the single-event mode, TRG should turn to L once RDY becomes L. Memory readout is performed for each of the 16 taps in parallel. SYNC is an output signal to indicate the beginning of the cycle of the shutter patterns. SYNC is used to trigger the emission of pulsed light e.g., in time-of-flight range imaging applications. To control the number of cycles or exposure time in multi-exposure mode, users are to count the number of SYNC pulses. When the count reaches an expected value, TRG should turn to L.

Here, the operation for a tap is described. To speed up the pattern readout and reduce power consumption, the shutter pattern is first transferred to the 8-bit shift register for every 8 bits and is then output bit by bit. The address of the memory is specified by ADR[4:0]. The length of the shutter pattern is given by SHLN[5:0]. When FETCH is asserted, the 8-bit output from the memory is fetched by the shift register. Otherwise, the content of the shift register is shifted by one bit. This configuration minimizes the number of circuits operating at the highest frequency, and is suitable for increasing the operational frequency of the shutter controller. Operation at 1 GHz was verified by simulation.

If TRG turns to L, RDY and ADR go back to H and 0, respectively, after the shutter patterns for the on-going cycle are read out. Next, the compressed signals stored in the storage diodes are read out in the same manner as in ordinary CMOS image sensors. 

## 4. Experimental Results

### 4.1. Basic Characteristics

The proposed image sensor was fabricated in a 0.11-micrometer CMOS image sensor process. A performance table and chip microphotograph are shown in Table 1 and Figure 5, respectively. The shown power consumption is for the whole chip. Table 2 summarizes the measured characteristics of the fabricated image sensor. The read noise and dark current are relatively high as the charge modulators store the charges in the floating diffusions.

Figure 6 shows the measured sensor responses for non-compression with sliding time windows and 8× compression with random shutters. To implement non-compression filming, tap-4 was used as a charge drain. Therefore, the number of time windows became 12. The random shutters were designed by the method described in [8]. The shutter patterns are shown in Figure 1. In measurement, a short pulse semiconductor laser (Tama Electric Inc., Hamamatsu, Japan, OPG-1000PL, wavelength of 445 nm, pulse width of <80 ps) and a delay controller (Stanford Research Systems, DG645) were used. It was found that the shortest time window (~3.3 ns) was resolved.

### 4.2. Single-Event Filming

As a single-event filming demonstration, a laser-induced plasma in the laser processing of a metal plate (SPCC) was captured. Figure 7 shows the experimental setup and captured and reproduced images. The burst frame rate was 303 Mfps. The shutters shown in Figure 1 were applied to perform compressive sensing with a compression ratio of 8×. A pulsed laser (Hamamatsu, L12968, wavelength of 1064 nm, pulse width of 0.5–2 ns, pulse energy of 10 mJ) was used. The laser emission and image capturing were triggered by a pulse from a function generator. Because the laser emission timing fluctuated due to the nature of the laser, the image sensor repeated exposure cycles and stopped when the plasma emission was detected by an external photodetector (Menlo Systems, FPD 310-FV). An ordinary commercial objective lens was placed in front of the image sensor. As shown in Figure 7b, from the four mosaic images captured at once, the number of which corresponds to that of the total taps in a macropixel, 32 sequential images were reproduced using the TV minimization method [30]. Note that the dimension of the TV was extended from two to three (x, y, and t). In the image reproduction, the measured temporal sensor response shown in Figure 6b and the PSF were incorporated as the matrix ***A*** in Equation (1). The number of pixels of the reproduced image was 188 × 212, which was the same as that of the subpixels. The generation and dispersion of the plasma were observed.

### 4.3. Transient Imaging of Light

Transient imaging is referred to as an emerging imaging modality in which the propagation of short light pulses in a scene is captured like a slow-motion film. Light emission and exposure were repeated multiple times to integrate the signal with the assumption that the scene was static while one frame was captured. A two-dimensional array of SPAD for the direct-type TOF (for example, see [31]) can perform transient imaging because a complete temporal waveform of the reflected light is recorded at each pixel. In the field of computational photography, the implementation of transient imaging has been attained with image sensors for the indirect-type TOF, since they are more cost effective or have more pixels compared with those for the direct-type TOF [32,33,34]. However, the techniques in the literature need multiple images to reproduce the time-sequential images, because the additional scanning of the light emission timing or modulation frequency of the light source is necessary. Therefore, they are not suitable for dynamic scenes. By contrast, our method can perform transient imaging in a single shot without any scanning. The reproduced temporally sequential images are similar to the histograms obtained in direct-type TOF depth imaging [35]. Although the charge modulators were developed for the indirect-type TOF, the proposed scheme enables us to perform quasi-direct-type TOF with simpler pixels and image sensor structure. Two different experiments were conducted: (1) the extension of the measurable range and (2) the decomposition of multiple reflections. 

In the first experiment, by using compressive sensing, the measurable range was doubled, with a compression ratio of 8× in terms of the pixel count. Because the number of taps in the macropixel is 16 and the frame rate is 303 Mfps, the depth resolution for one frame and the measurable depth range in time-of-flight depth imaging are about 0.5 m and 8 m, respectively. The range was extended to 16 m with compressive sensing. In the measurement, four objects were placed 4 m, 6 m, 8 m, and 13 m away from the camera. A semiconductor pulsed laser (Tama Electric Inc., Hamamatsu, Japan, LDB-320, wavelength of 660 nm) emitted light pulses with a full width of ~7 ns, triggered by SYNC. From the captured images for the 391,807 light pulses, a series of 32 time-resolved images was reproduced (Figure 8b). 100 images were averaged to improve the signal-to-noise ratio. Each object was observed at the frames that corresponded to their distances. An ordinary single-aperture imaging lens (Edmund Optics, C series VIS-NIR fixed focal length lens, focal length of 16 mm, f-number/1.4) was used. The time of flight was calculated at each pixel by the temporal centroid of each intensity peak from the reproduced images in Figure 8b. This was then converted to the depth based on d=τc/2, where d,τ, and c are depth, time of flight, and speed of light, respectively. Figure 8c shows the reproduced depth map in the point cloud representation.

In the second experiment, interference light was introduced by a weak diffusive plastic sheet placed in front of the letters “SU” (Figure 9a). Such a situation is referred to as multipath interference [36], where multiple reflections are on the same line of sight. The distance between them was 3.3 m. After reproduction, two reflections from the two objects were separated, as shown in Figure 9b. The separation was 3.465 m, based on the detected frame positions. The light source and optical and electrical setup were the same as those in the first experiment. While it was difficult to decompose mixed multiple reflections in the conventional indirect TOF based on charge modulators, compressive sensing solved this problem. Figure 9c shows a point cloud of the reproduced depth map from the captured images. It is shown that the two objects were separated, although there was some crosstalk between them. The crosstalk could have been caused by the difference between the measured and real sensor responses. Further investigation is required to reduce it.

## 5. Discussion

In this paper, the concept, implementation, and demonstration of the UHS CMOS image sensor based on computational photography were presented. The image acquisition scheme is based on compressive sensing, and signal compression is performed in the charge domain in pixels. Therefore, our method is free from the limitations from which conventional CCD and CMOS UHS image sensors suffer. The frame rate of 303 Mfps was achieved, and further improvement in the performance will be possible, as discussed below. From the viewpoint of applications, the compatibility with conventional single-aperture optics is crucial, as the developed image sensor can be combined with a variety of conventional optical systems. The dual-mode operation for single-event filming and multi-exposure filming is also important, since the same cost-effective image sensor can serve both in consumer markets and the sciences. It is expected that new applications will be explored based on these features. In autonomous driving applications, SPAD-based TOF image sensors are regarded as the most promising. However, as demonstrated in Section 4.3, the developed image sensor provides time-resolved images similar to those obtained by SPAD-based TOF image sensors, although the pixel structure is much simpler than that of the SPAD. Our imaging scheme and image sensor architecture can be an option to implement a pseudo-direct-type TOF.

To improve the image sensor’s performance, several issues on skew, observation period, reproductions algorithms, and response time for long wavelengths are discussed below. 

For UHS image sensors, accuracy in time and frame rate (or time resolution) are of significant importance. The accuracy in time means the difference in the observation timings among the pixels and taps, which is defined by the skew of the shutter signals. Figure 10 shows the measured skew histogram and distribution for the rising edge of the same time window timing, whose width should ideally be zero. The finite distribution is caused by the fact that the shutter timings are different among the taps or pixels. Figure 10a is the histogram of the skew among all the pixels and taps. The distribution is almost the same at the shortest exposure duration, namely, 3.3 ns for 303 Mfps. Because the temporal response of every tap and pixel is measured and incorporated in inverse problem solving, the skew is cancelled. However, the skew should be smaller to simplify the post-processing. Figure 10b shows the distribution of the skew for the tap-1 of pixel-1. Because the charge modulator drivers are situated only at the bottom of the image, the skew becomes larger in the top. Probably due to the voltage drop of the power and ground lines of the charge modulator drivers, the skew in the center in horizontal becomes larger. Note that our image sensor has optical black pixels in the right of the image, which is not depicted in Figure 10b. Therefore, the skew distribution appears asymmetric.

To reduce the skew, it is effective to place the charge modulator drivers at both ends of the pixel array (top and bottom). Furthermore, local drivers embedded in the pixel array can alleviate the skew by virtually reducing the load of the charge modulator drivers [37]. The local driver is also effective at improving the frame rate because it can implement shorter time windows. However, the local drivers will consume some areas in the pixel array and reduce the fill factor. It is very effective to adopt the stacking technology [38] to implement the local drivers without degrading the fill factor. High fill factors enable us to use small f-number lenses, which is very important, since UHS imaging is mostly photon-hungry. Stacking technology also allows elaborately designed clock trees to provide the charge modulation signals to all the pixels at the same time.

To increase the number of frames for longer observation time or over longer ranges in TOF, the total number of taps in a macro-pixel should be increased. Namely, we need more subpixels in a macro-pixel and more taps in each subpixel. Increasing the compression ratio is also important because it can extend the observation time or depth range. Although a compression ratio of 8× was used in this study, a compression ratio of 32 has been demonstrated in [9,39] based on dictionary or learning-based methods. For specialized applications, such as TOF depth imaging, these methods can be efficient. For real-time applications, it is necessary to speed up the inverse problem-solving. Iteration-based methods are slow, and it is difficult to exactly estimate the signal processing time beforehand. Recently, deep-learning-based methods have been investigated in compressive video, which does not require any iteration [40].

In this paper, a wavelength of 660 nm was used in the transient imaging of reflected light because the response of the photodiode was relatively slow for near-infrared wavelengths. However, a fast near-infrared response is necessary in TOF depth imaging. The response of the charge modulator is composed of those of the photodiode and charge modulator itself. The modulator response is almost independent of the wavelength, while the photodiode response is somewhat dependent on it due to the wavelength dependency of the penetration depth of light in silicon. It was demonstrated that backside biasing was effective at improving the response speed of a photodiode [41]. 

## 6. Conclusions

The design and demonstration of an ultra-high-speed computational CMOS image sensor based on charge-domain temporally compressive sensing with a burst frame rate of 303 Mfps, which is the fastest among solid-state image sensors, to our knowledge, were described. To achieve such a high frame rate, the charge modulator drivers are separated into four blocks for each of the four subpixels to reduce the peak driving current on the shared power and ground lines. The pixel array is composed of macropixels with 2 × 2 4-tap subpixels. The image sensor was fabricated in 0.11-micrometer CMOS image sensor technology.

This image sensor is compatible with ordinary single-aperture optics and can operate in dual modes, such as single-event filming mode or multi-exposure transient imaging mode, selectively, so that it can be combined with any conventional optics and can be applied to a variety of applications. 

In the experiments, 32 images were reproduced from 4 compressed mosaic images with a compression ratio of 8×. As an example of single-event filming, a laser-induced plasma was observed at 303 Mfps. Transient imaging in a single shot and depth map acquisition based on the time of flight were also demonstrated. The depth range was doubled by compressive sensing. In addition, multiple reflections were separated, as with direct-type TOF image sensors, although our image sensor utilized the charge modulator designed for the indirect-type TOF.

To realize higher frame rates with low skew, local charge modulator drivers should be integrated in the pixel array. Stacking technology is desired to implement the local drivers at high fill factors. For TOF applications, the temporal response of charge modulators for near-infrared wavelengths should be improved. The development of image reproduction algorithms based on deep neural networks is also required to increase the compression ratio and to reduce the image reproduction time.

## Figures and Tables

**Figure 1 sensors-22-01953-f001:**
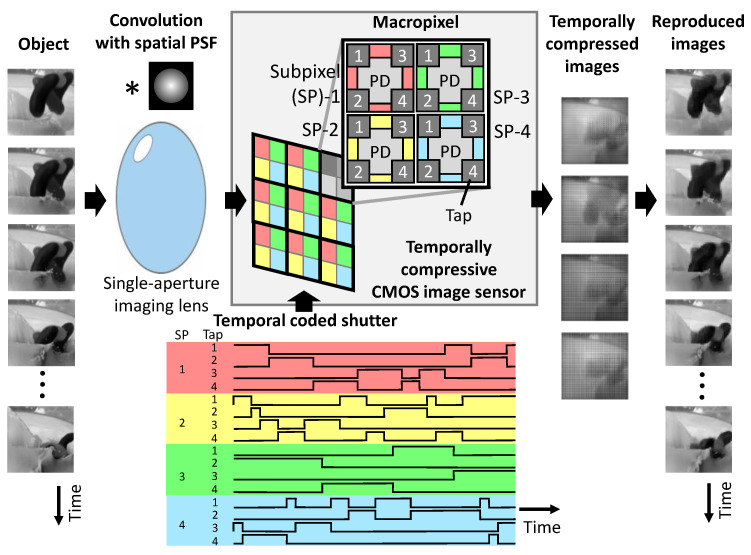
Flow of image acquisition and reproduction. * is the convolution operator.

**Figure 2 sensors-22-01953-f002:**
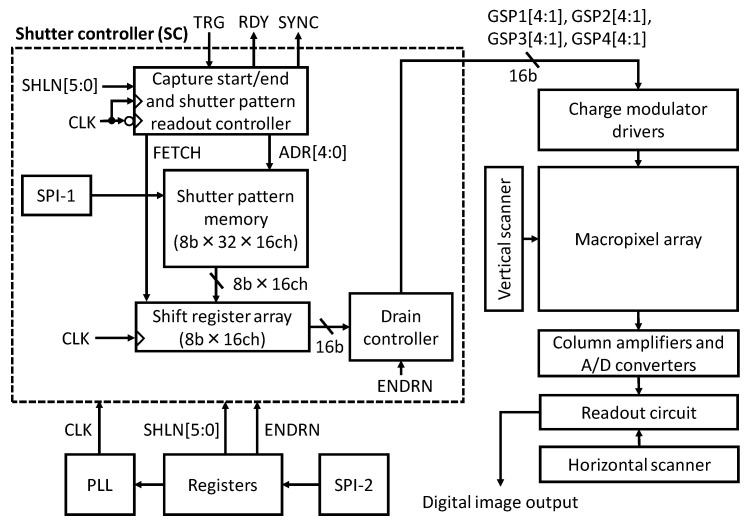
Image sensor architecture.

**Figure 3 sensors-22-01953-f003:**
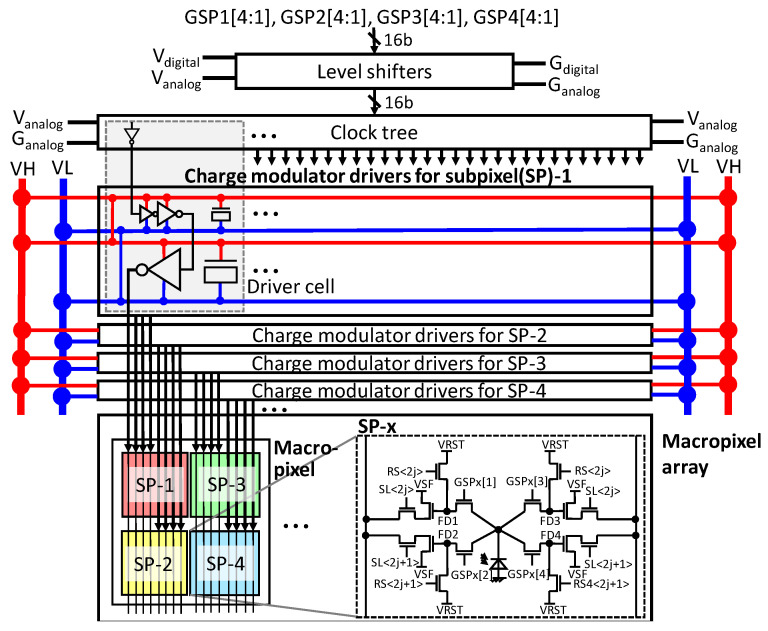
Charge modulator drivers and pixel array.

**Figure 4 sensors-22-01953-f004:**
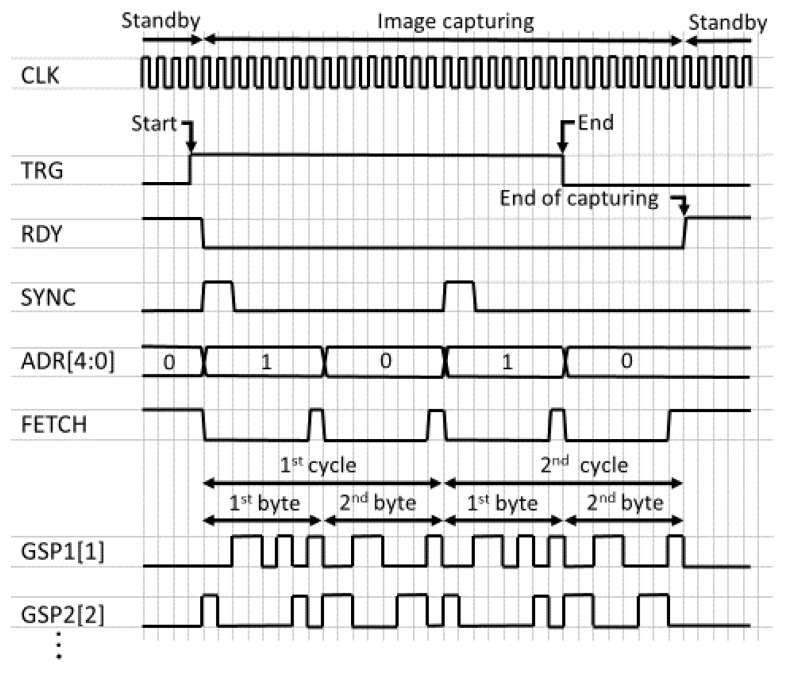
Timing chart of shutter controller for filming. The number of repetitions is two in this figure.

**Figure 5 sensors-22-01953-f005:**
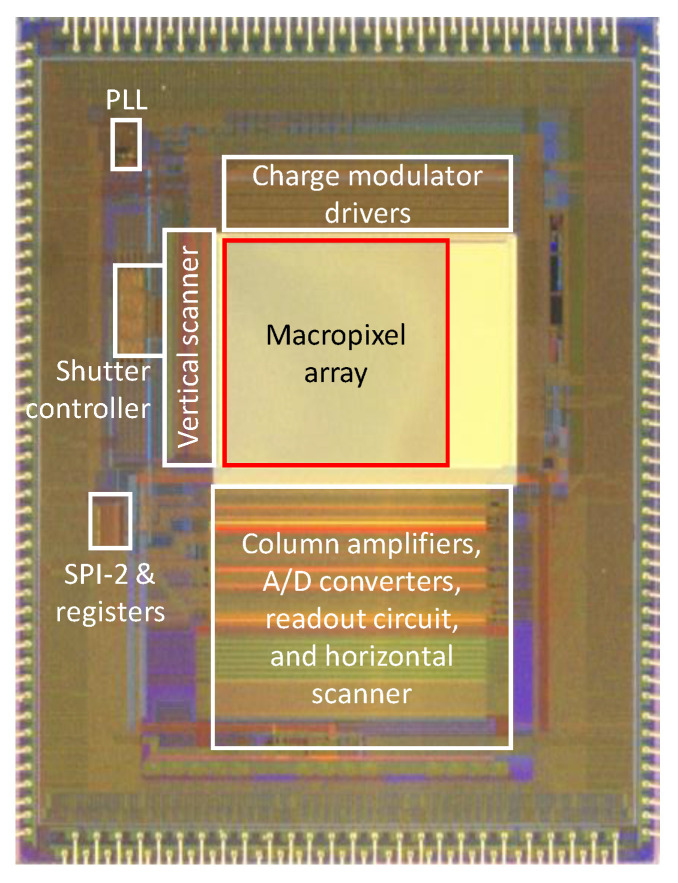
Chip microphotograph.

**Figure 6 sensors-22-01953-f006:**
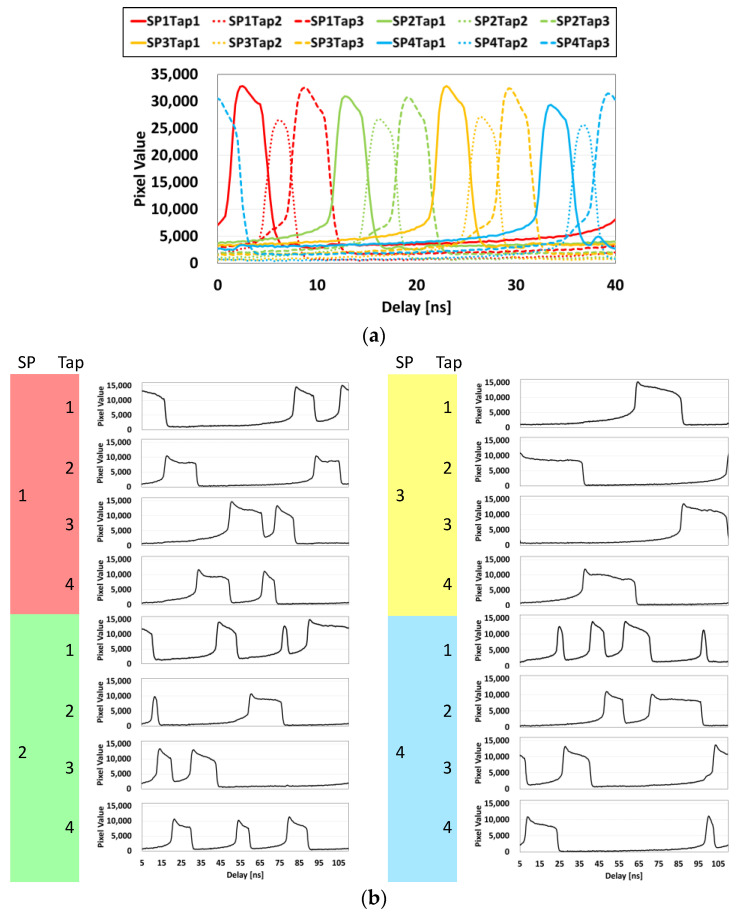
Sensor responses measured with a short pulse semiconductor laser (λ = 445 nm, pulse width < 80 ps). (**a**) Non-compression with sliding time windows. (**b**) 8× compression with random shutters. SP: subpixel.

**Figure 7 sensors-22-01953-f007:**
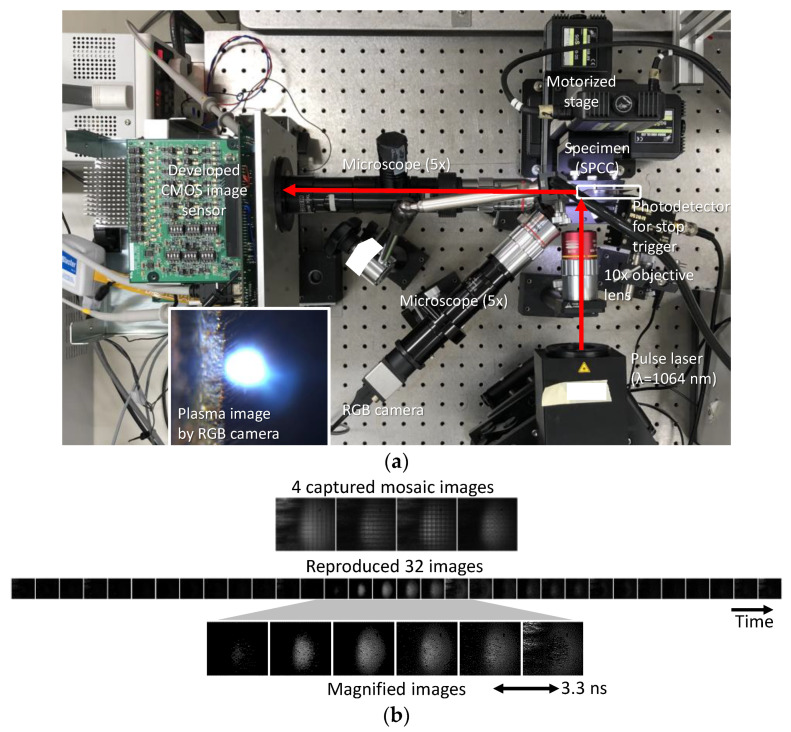
Single-event filming of plasma emission. (**a**) Experimental setup. (**b**) Captured and reproduced images.

**Figure 8 sensors-22-01953-f008:**
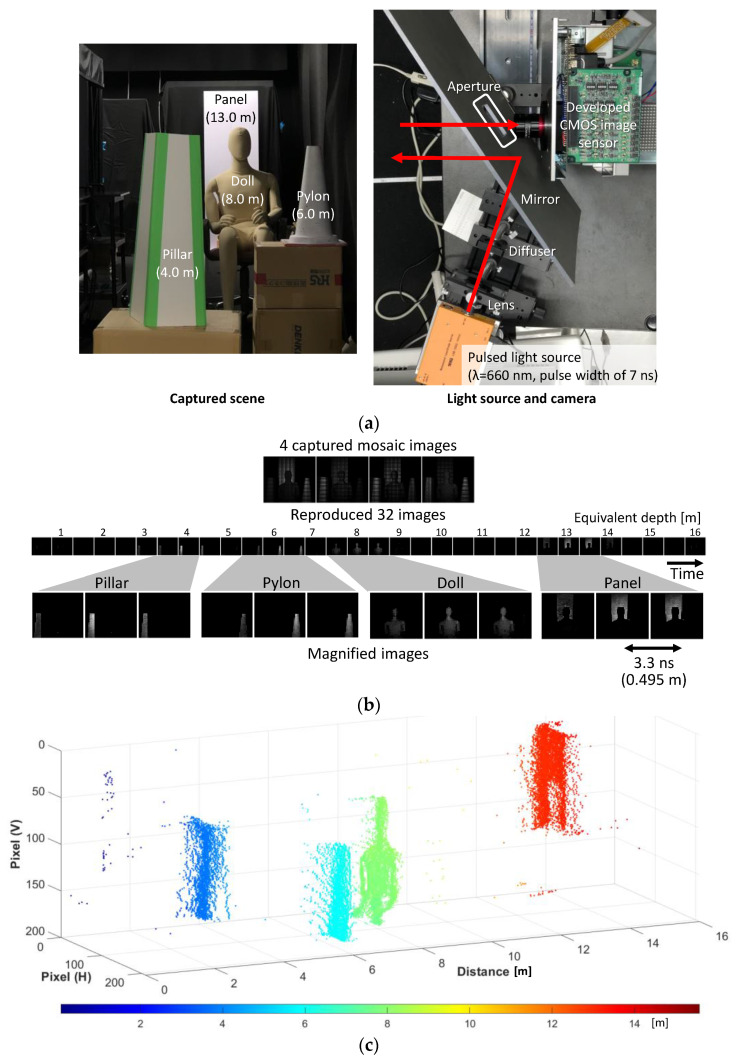
Multi-exposure transient imaging of reflected light. The extended measurable range was doubled with compressive sensing. (**a**) Experimental setup. (**b**) Captured and reproduced images. (**c**) Point cloud representation of the depth map calculated from the reproduced images.

**Figure 9 sensors-22-01953-f009:**
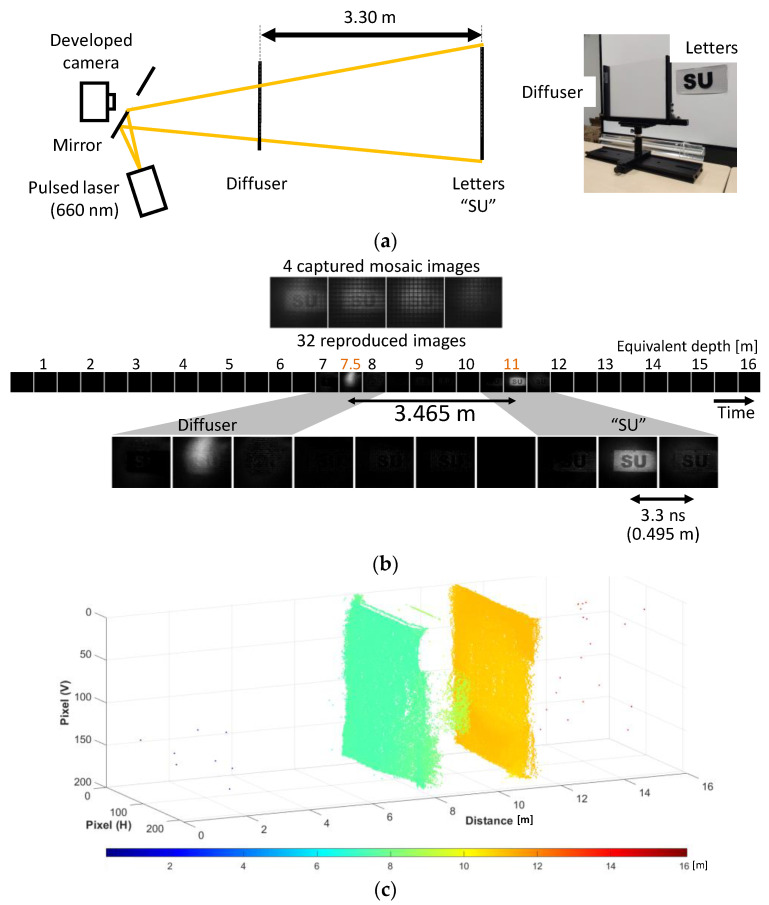
Multi-exposure transient imaging of multiple light reflections. Two reflections on the same line of sight were decomposed. (**a**) Experimental setup. (**b**) Captured and reproduced images. (**c**) Point cloud representation of the depth map calculated from the reproduced images.

**Figure 10 sensors-22-01953-f010:**
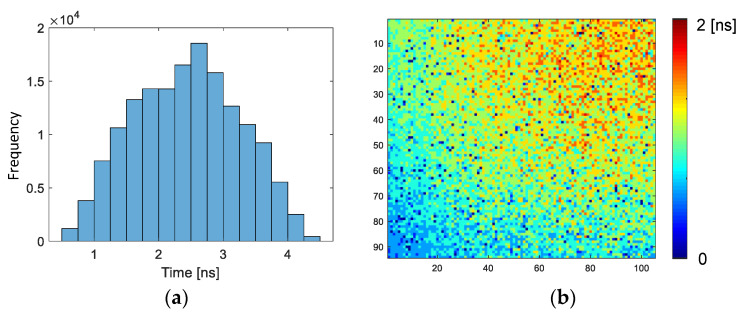
(**a**) Skew histogram for all taps and pixels and (**b**) the distribution of skew for tap-1 of subpixel-1.

**Table 1 sensors-22-01953-t001:** Comparison of specifications and performance.

	This Work	ISSCC’15 [7]	IISW’19 [6]	MDPI Sensors [5]	ISSCC’12 [4]
Technology	0.11 μm CMOS FSI	0.11 μm CMOS FSI	0.18 μm CMOS FSI	0.13 μm CMOS/CCD BSI	0.18 μm CMOS FSI
Chip size	7.0 mm^H^ × 9.3 mm^V^	7.0 mm^H^ × 9.3 mm^V^	4.8 mm^H^ × 4.8 mm^V^	-	15 mm^H^ × 24 mm^V^
(Macro)pixel size	22.4 μm^H^ × 22.4 μm^V^	11.2 μm^H^ × 5.6 μm^V^	70 μm^H^ × 35 μm^V^	12.73 μm^H^ × 12.73 μm^V^	32 μm^H^ × 32 μm^V^
Effective (sub)pixel count	212^H^ × 188^V^	320^H^ × 324^V^	50^H^ × 108^V^	576^H^ × 512^V^	400^H^ × 250^V^
(Sub)pixel count per macro pixel/aperture	2^H^ × 2^V^	5^H^ × 3^V^	-	-	-
Tap count per (sub)pixel/aperture	4	1 + drain	-	-	-
Maximum shutter length	256b	128b	-	-	-
Number of frames in burst operation	12@1×, 32@8×	15@1×, 30@2×	368 (in-pixel), 184 (off-pixel)	10	248
Maximum burst frame rate	303 Mfps	200 Mfps	100 Mfps (in-pixel), 125 Mfps (off-pixel)	100 Mfps	20 Mfps
Image readout frame rate	21 fps	22 fps	N/A	-	15 kfps
Power consumption	2.8 W	1.62 W	N/A	-	24 W
Multiple exposure	Yes	Yes	No	No	No
Compatibility with normal optics	Yes	No	Yes	Yes	Yes

**Table 2 sensors-22-01953-t002:** Measured characteristics.

Conversion gain	32.5 μV/e^−^
Read noise	85 e^−^_rms_
Full-well capacity	33,000 e^−^
Dark current (average)	3043 e^−^/s@room temperature
Quantum efficiency	40.6%@660 nm

## Data Availability

Not acceptable.

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
