# Peer review of "A Dual-Mode 303-Megaframes-per-Second Charge-Domain Time-Compressive Computational CMOS Image Sensor"

_sensors, 2022, doi:10.3390/s22051953_

Round 1

Reviewer 1 Report

The work could be interesting. The overall presentation and discussion are not well presented. Significant improvement is expected as list below.

  1. More compressive sensing including temporal and spatial compression needs to be reviewed and discussed e.g. C Tang, etc., Smart compressed sensing for online evaluation of CFRP structure integrity IEEE Transactions on Industrial Electronics 64 (12), 9608-9617, 2017;
  2. The discussion of temporally compressive sensing, transient imaging and stacking technology are not well linked. The novelty should be highlighted. Further refinement in sections 2.2 and 5 are expected;
  3. The conclusion and further work should highlight the major contribution and discussion of future work. 

Author Response

Thank you very much for your valuable suggestions. I tried to improve my manuscript based on your advice. I hope that the revision is suitable for publication based on your standard.

> 1. More compressive sensing including temporal and spatial compression needs to be reviewed and discussed e.g. C Tang, etc., Smart compressed sensing for online evaluation of CFRP structure integrity IEEE Transactions on Industrial Electronics 64 (12), 9608-9617, 2017;

I have added more articles in 2.2 as well as the article you suggested.

> 2. The discussion of temporally compressive sensing, transient imaging and stacking technology are not well linked. The novelty should be highlighted. Further refinement in sections 2.2 and 5 are expected;

I have added more explanation on the transient imaging and depth data by the point cloud representation in 4.3 to clarify the relationship between the transient imaging and time-of-flight depth imaging.

I have added the data on the skew of our image sensor to explain the necessity of the stacking technology. And, most parts of Sec. 5 were rewritten to explain the necessity of stacking technology.

> 3. The conclusion and further work should highlight the major contribution and discussion of future work. 

The conclusion has been rewritten to highlight the novelty, major contributions, and future work.

Others:

To highlight the technological keys, explanations have been added to the last paragraph of Sec. 1 and the first paragraphs of 3.1 and 3.2.

Reviewer 2 Report

The paper is well written and presents a new type of ultra-high-speed image sensor.

Though there are nice results from the different application examples, the measurement part in Chapter 4.1 could be more extensive to show dynamic range and other basic image sensor performance.

Author Response

Thank you very much for your comments.

I have added the characteristics table of our image sensor, depth maps in the point cloud representation, and a skew map.

Round 2

Reviewer 1 Report

The improvement is reasonably good. Further refinement is still expected as list below.

  1. The title could be shortened and refined;
  2. The logic link and correspondence of the title, abstract, major contribution and conclusion need to be improved;
  3. Overall discussion of the contribution beyond implementation is required.

Author Response

Thank you so much for giving me advisable suggestions. I have revised my manuscript based on your comments.

> 1. The title could be shortened and refined;

I have refined the title.

> 2. The logic link and correspondence of the title, abstract, major contribution, and conclusion need to be improved;

I tried to improve the consistency among the title, abstract, major contribution, and conclusion.

> 3. Overall discussion of the contribution beyond implementation is required.

I have added the discussion in the first part of the discussion section.

Others.

I have corrected the reference numbers.

I have corrected Fig. 1. The background color on the timing chart was a bit wrong.

I have inserted some sentences to show our contribution more clearly.